# Clinico-Genomic Analysis Reveals Mutations Associated with COVID-19 Disease Severity: Possible Modulation by RNA Structure

**DOI:** 10.3390/pathogens10091109

**Published:** 2021-08-31

**Authors:** Priyanka Mehta, Shanmukh Alle, Anusha Chaturvedi, Aparna Swaminathan, Sheeba Saifi, Ranjeet Maurya, Partha Chattopadhyay, Priti Devi, Ruchi Chauhan, Akshay Kanakan, Janani Srinivasa Vasudevan, Ramanathan Sethuraman, Subramanian Chidambaram, Mashrin Srivastava, Avinash Chakravarthi, Johnny Jacob, Madhuri Namagiri, Varma Konala, Sujeet Jha, U. Deva Priyakumar, P. K. Vinod, Rajesh Pandey

**Affiliations:** 1INtegrative GENomics of HOst-PathogEn (INGEN-HOPE) Laboratory, CSIR-Institute of Genomics and Integrative Biology (CSIR-IGIB), Mall Road, Delhi 110017, India; priyanka.m@igib.in (P.M.); aparnamurali10@gmail.com (A.S.); saifi.sheeeba@yahoo.in (S.S.); ranjeet.maurya@igib.in (R.M.); partha.c@igib.in (P.C.); priti.devi@igib.in (P.D.); akshay.kanakan@igib.in (A.K.); jananisv@igib.in (J.S.V.); 2Center for Computational Natural Sciences and Bioinformatics, International Institute of Information Technology, Hyderabad 500032, India; shanmukh.alle@research.iiit.ac.in (S.A.); anushachaturvedi7@gmail.com (A.C.); ruchi.chauhan@research.iiit.ac.in (R.C.); 3Academy of Scientific and Innovative Research (AcSIR), Ghaziabad 201002, India; 4Intel Technology India Private Limited, Bangalore 530103, India; ramanathan.sethuraman@intel.com (R.S.); subramanian.c@intel.com (S.C.); mashrin.srivastava@intel.com (M.S.); avinash.chakravarthi@intel.com (A.C.); johnny.jacob@intel.com (J.J.); madhuri.namagiri@intel.com (M.N.); varma.s.konala@intel.com (V.K.); 5Max Super Speciality Hospital (A Unit of Devki Devi Foundation), Max Healthcare, Delhi 110017, India; sujeet.jha@maxhealthcare.com

**Keywords:** COVID-19, clinico-genomic, mutation, RNA structure, disease severity, clinical outcome, integrative analysis

## Abstract

Severe acute respiratory syndrome coronavirus 2 (SARS-CoV-2) manifests a broad spectrum of clinical presentations, varying in severity from asymptomatic to mortality. As the viral infection spread, it evolved and developed into many variants of concern. Understanding the impact of mutations in the SARS-CoV-2 genome on the clinical phenotype and associated co-morbidities is important for treatment and preventionas the pandemic progresses. Based on the mild, moderate, and severe clinical phenotypes, we analyzed the possible association between both, the clinical sub-phenotypes and genomic mutations with respect to the severity and outcome of the patients. We found a significant association between the requirement of respiratory support and co-morbidities. We also identified six SARS-CoV-2 genome mutations that were significantly correlated with severity and mortality in our cohort. We examined structural alterations at the RNA and protein levels as a result of three of these mutations: A26194T, T28854T, and C25611A, present in the Orf3a and N protein. The RNA secondary structure change due to the above mutations can be one of the modulators of the disease outcome. Our findings highlight the importance of integrative analysis in which clinical and genetic components of the disease are co-analyzed. In combination with genomic surveillance, the clinical outcome-associated mutations could help identify individuals for priority medical support.

## 1. Introduction

Since the emergence of SARS-CoV-2 in late 2019, it has diversified into numerous different variants, which have been reported and associated with an increase in infection rates worldwide. RNA viruses are observed to have a significantly higher mutation rate as compared with DNA viruses [1]. However, coronaviruses tend to have a lower mutation rate comparatively because they encode proteins that rectify the incorrect nucleotide incorporation during the time of replication. These regularly occurring mutations cause changes in virus genome architecture and functional characteristics, which holds the potential to alter the viral pathogenicity [2]. Thus, certain mutations are associated with mild disease severity, allowing the virus to spread and maintain the outbreak. At the same time, the presence of other mutations leads to severe disease symptoms and resultant adverse impacts on the patient/s. 

SARS-CoV-2 enters the host cell by binding to the host ACE2 receptor via spike protein and the virus entry subsequently results in COVID-19 infection. Patients have varied clinical manifestations and the most common symptoms observed are fever, cough, anorexia, dyspnea, sputum production, and myalgia, as described by CDC [3]. Interstitial and alveolar pneumonia is the primary clinical manifestation reported among the COVID-19 patients [4]. Cardiovascular complications include acute pericarditis, left ventricular dysfunction, and acute myocardial injury [5].Viral inflammation can also cause myocyte injury, thereby increasing the challenge for a weak heart to function [6]. Podocytes and proximal tubular epithelial cells in the kidney are found to have a higher number of ACE2 receptors and can serve as a target for SARS-CoV-2 infection [7]. Elevated liver enzymes and liver injury were found to be correlated with increased time of hospitalization; however, they were not associated with the risk of death [8]. Furthermore, neurological symptoms such as encephalopathy, encephalitis, seizures, cerebrovascular events, acute polyneuropathy, headache, hypogeusia, and hyposmia have also been reported [9]. In severe cases, there can be a dramatic elevation of D-dimer values and low platelet count [10].

COVID-19 has a wide range of clinical manifestations, ranging from asymptomatic infection to severe symptoms leading to mortality. Elucidating what causes COVID-19 patients to have a wide spectrum of clinical symptoms is the subject of an ongoing investigation from both the viral and host perspectives. Understanding the impact of mutations in the SARS-CoV-2 genome on the clinical phenotype and associated comorbidities is critical for the treatment and containment of the virus as the pandemic progresses. So far, studies have suggested that older age, male gender, the existence of comorbidities such as diabetes and hypertension, as well as demographic and clinical variables, have a role in increased susceptibility towards having a severe form of the COVID-19 disease [11,12]. 

Various diagnostics methods, apart from conventional methods, have been developed to ascertain SARS-CoV-2 infected patients to effectively limit the risk of virus transmission [13]. This includes CRISPR-Cas9 based detection of the wild-type SARS-CoV-2 in general as well as specific variants, for example Alpha variant using RAY (Rapid variant AssaY). It is a paper-strip based platform to identify mutational signatures of the coronavirus variants in a sample. Genome sequencing-based identification of the signature mutations associated with the VOC is again the enabler towards development of such diagnostic methods [14].Genome sequencing also plays a pivotal role in controlling the progression of the pandemic by giving us insights into the viral genome architecture and dynamics. This includes the earlier incidences of the viral entering a geographical region. The NGS-enabled genome sequencing has been instrumental in identifying the viral clades, lineages, and the role of non-synonymous mutations in altering the protein structure [15]. SARS-CoV-2 genome analysis can help us identify novel variants, reinfection cases, and also track the source of infection [16].Genomic surveillance has been instrumental to conclusively prove the incidences of re-infection. The identified mutations have been studied in detail for their role in immune escape through functional studies to enhance and elucidate the mechanism [17]. Genome sequencing coupled with serological studies have been used for genomic characterization and epidemiology of the emerging SARS-CoV-2 variants vis-a-vis multiple waves of infection. With vaccination breakthroughs being reported worldwide, genomic surveillance has been helpful to detect the variants and identify the mutations associated with the immune escape [18]. A study reported by Luke et al. demonstrated how rapid genomic sequencing along with epidemiological data was used to investigate healthcare workers-associated infections and their transmission network [19]. All of the insights highlighted above have been facilitated by genome sequencing of SARS-CoV-2. However, at the same time, it has been possible due to the background clinical and epidemiological information from hospital-admitted patients as well as population-level genomic surveillance. Diagnostic tools and NGS have been of great assistance inthe development of vaccines and therapeutics, by assessing the pattern of global spread, and genetic heterogeneity while the pandemic is ongoing [20]. However, an understanding based on the sequence analysis of evolving viral genomes has crucial implications for both strategic planning in disease prevention, disease progression, and the development of vaccines and therapeutics, while the pandemic is ongoing.

Many studies have highlighted the possibility of the functional role of various SARS-CoV-2 genetic variations and their correlation with disease severity. A study by Nagy etal.reported mutations in the ORF3a and NSP7 protein to be associated with severe outcomes [21]. Young et al. discovered a ∆382 deletion in the ORF8 gene from a cohort of SARS-CoV-2 infections in Singapore, and it was found to be correlated with mild cases [22]. A study on 141 clinical samples from Hubei province of China reported an inframe deletion, ∆500–532 located in the N terminal region of the NSP1 gene that was found in more than 20% of the genome sequences. Notably, the deletion ∆500–532 was seen to be associated with a lower level of IFN-β in the serum isolated from the patient samples and also had a higher Ct value in the RT-PCR [23]. Another study by Esper etal.demonstrated clade-wise association of variants with clinical severity. Patients infected with clade V virus strains were presented with overall higher mortality rates comparatively. At the same time, increased variants in the ORF3a were associated with decreased hospitalization frequency, whereas increased variants in Spike were associated with increased survival when hospitalization was required [24].

In the present study, we comprehensively analyzed the SARS-CoV-2 genomes of 196 hospitalized patients in India. We conducted an integrative study of the evolving SARS-CoV-2, focusing on the mutation analysis and its impact on patient’s disease severity and clinical outcomes. We were able to successfully identify six mutations that show significant association with disease severity and mortality. To further explore its plausible functional repertoire, selective mutations were found to modulate the RNA structure. The findings of the study highlight the importance of integrative understanding through a combination of genome sequencing, mutation analysis, effect on the RNA structure, and its possible role in disease severity and clinical outcome.

## 2. Results

### 2.1. Demographics and Diversity of Clinical Features and Severity in COVID-19 Patients

The patients (N = 196) included in the study exhibited shortness of breath, body ache, fever, nausea and were admitted to intensive care units (ICU) of the MAX hospital in Delhi, India. Of these, the majority of the patients had frequently reported fever (76%) and cough (47%). Patients with co-morbidities such as diabetes (27%), hypertension (59%), heart ailments (23%), and hypothyroidism (9%) became relatively more severely ill, even when intervened through respiratory support and medical support with antivirals (18%), hydroxychloroquine (HCQ) (23%), and steroids (23%). For clinical comparison, the patients were grouped into recovered (R) (*n* = 174; female/male (*n*):55/122) and dead (D) (*n* = 22; female/male (*n*):6/16). The median age of recovered patients was found to be 52 years (32–64) and death patients to be 65 years (55–70), which was significant (*p* < 0.001) (Table 1). Between the R and D groups, there was a significant difference (*p* < 0.05) in favor of the D group for the onset of certain symptoms, which includes fever (*p* = 0.04), sore throat (*p* = 0.018), difficulty in breathing (*p* < 0.001), co-morbidities [hypertension (*p* < 0.001), and heart conditions (*p* = 0.0025)]. Significant differences (*p* < 0.05) between these two groups (in favor of the D group) were also observed in respiratory support requirements (*p* < 0.001). A subset of patients under respiratory support also required ventilator support. On comparing these patients between R and D groups, a significant difference was observed between the duration of ventilator support required (*p* = 0.034). The treatment with HCQ was also found to be statistically significant between the R and D groups (*p* = 0.006) (Table 1).

To further understand the spectrum of clinical phenotypes, the R group patients were subdivided into mild (blue) and moderate (orange) severity classes and the D was grouped as severe (red) class (Figure 1). The mild and moderate classes were distinguished based on the non-respiratory support and requirement of respiratory support, respectively. Out of the 174 recovered patients, 98 patients (56%) did not require respiratory support and thus were classified as a mild category. The remaining 76 patients in the moderate class required respiratory support. It was observed that patients in the moderate class stayed on ventilator support for an average of 13 days with one patient staying on a ventilator for more than a month, while patients in the severe class had an average stay of sixdays (Figure 1A). The spider web plot represents the distribution of patient samples for co-morbidities and symptoms concerning the three severity classes (Figure 1B). Diabetes and hypertension were predominant in the moderate class patients. Diabetes and hypothyroidism were seen in some of the mild category patients. It was observed that patients without co-morbidities had a higher frequency for mild class patients in the study. However, no statistical difference was observed between the R and D group of patients without co-morbidities (*p* value = 0.065), while for severe class patients, hypertension was the common co-morbid condition. The spectrum of symptoms was diverse for all three classes. Probably as expected, breathing distress was the most common symptom for the severe category, but importantly, the mild category also showed this trend. However, the moderate class showed sore throat, body ache, and general weakness as common symptoms (Figure 1B). 

### 2.2. Phylogenetic and Mutation Variation of SARS-CoV-2

Using the genome sequencing of the SARS-CoV-2 isolated from all the patients, viral clades and lineages were analyzed. We identified three clades according to the Nextclade classification, i.e., 19A, 20A, and 20B. Clade 20A defined by C14408T (Nsp12/RdRp) and A23403G (S: D614G) was seen in 58% of the samples, clade 19A defined by positions 8782C (Nsp3) and 14408C (Nsp12/RdRp) was seen in 38% and 20B denoted by positions C3037T (Nsp3: 106F); A23403G (S: D614G); C14408T (Nsp12/RdRp: P4715L) and G28881A and G28882A (N: R203K) was seen in only 4% of the population (Figure 2A). Lineages B.1.36, B.6, and B.1 were the frequent lineages based on PANGOLIN classification (Figure 2B).

Based on the genomic sequences, top frequent mutations were identified (Table 2). Mutation P314L was seen in 75% of the patient samples, followed by D614G in 63.8% samples. C22444T, another spike mutation, was seen in 39.3% of the viral sequences. The Q57H mutation of ORF3a was seen in more than 50% of the isolates (Table 2). 

### 2.3. Association of Mutation with Disease Severity

Analysis towards the correlation of the observed mutation with disease severity showed 15 mutations to be correlated with both severity and mortality (Appendix A). Out of the 15 mutations, eight were common between both the sub-groups. This observation is expected as disease severity and mortality are highly correlated. The high correlation of a mutation with both severity and mortality reinforces its significance. We used more than one criterion to select the mutations with the aim to make it stringent. First, we selected all the mutations that had a *p*–value of <0.05, then we selected mutations that were common for bothseverity and mortality groups. Subsequently, based on the gene locus and amino acid change (synonymous vs. non-synonymous), we selected the mutations A26194T, C28854T, C25611A for further structural analysis (Figure 3). The mutations A26194T (*p* = 0.0023) and C25611A (*p* = 0.0158) were selected as they have a statistically significant correlation with disease severity. It can also be observed that the presence of the mutations correlated with a higher rate of severe cases. We found that 92% and 72% of the cases with the mutations A26194T and C25611A, respectively, were severe compared to an average of ~50%, which shows that the mutations are positively correlated with disease severity. 

The mutations A26194T (*p* = 0.0027) and C28854T (*p* = 0.0366) were selected as they have a statistically significant correlation with disease mortality. We observed that the presence of the mutations correlated with an increased rate of fatal cases. We found that 38% and 20% of the cases with the mutations A26194T and C28854T were fatal, compared to an average mortality rate of 10.5%, which shows that the mutations are positively correlated with mortality. 

Next, we queried whether the presence or absence of co-morbidity affects mutation association with disease severity and mortality. The analysis showed that mutation A26194T was found to be positively correlated with both severity and mortality (Appendix A). The mutation was found in all the patients with co-morbidities who developed severe COVID-19 infection, of which more than 40% cases were fatal. For patients without co-morbidities, mutation TG11082T in Orf1a was found to positively correlate with both severity and mortality (Appendix A). Incidentally, this mutation was present in around 5% of the patients with no co-morbid conditions.

### 2.4. Mutations Modulating the Structure 

We analyzed the positively correlated mutations associated with disease severity and mortality and their effect on the protein and RNA structure. We focused on three mutations: A26194T (T268S) in the Orf3a region, C25611A (synonymous mutation) in the Orf3a region, and C28854T (S194L) in the N protein. During the protein structure analysis, we observed a minor change in polarity due to the T268S (Appendix A) and S194L (Appendix A) with no significant change in the physicochemical properties. We further analyzed the secondary structure of the Orf3a protein, wherein we observed a conversion from sheets to coils at the site of mutation for T268S mutation (Figure 4). For S194L mutation, coils were turned into sheets (Figure 5). 

We also observed the protein disorder caused by mutations. Our analysis revealed that T268S increased the protein disorder from segment (244–267)to segment (244–272) in the Orf3a region, whereas S194L mutation did not have any effect on protein disorder.

To complement the protein structure analysis, we also analyzed the RNA secondary structure of SARS-CoV-2 genome sequence (NC_045512*), before and after the mutation in the Orf3a, and the N region. We considered NC_045512* for our analysis where we took the entire protein region as an input for the Orf3a and N region. For A26194U mutation in the Orf3a region (CDS Start:25393 and CDS End: 26220), we considered the entire Orf3a region as an input to RNAfold; we found that A at the 26194th site does not bind with any other residue and forms a 6nt loop with CAG repeats before the mutation takes place. However, after the mutation from A to U, A present at the 26141st site binds with the U at the 26194th site. This converts the stem loop into a stem loop structure, as depicted in Figure 6A,B. When we analyzed for C28854U (CDS Start: 28274 and CDSEnd: 29533) and C25611A (CDS Start:25393 and CDS End: 26220), the mutations led to a complete change of the secondary RNA structures and werenot limited to the site of mutation only (Appendix A). We also tried RNAStructure Web Server [25] to validate the secondary structures predicted by RNAFold and found the RNAfold and RNAstructure findings to be consistent with each other.

Additionally, we also compared the RNA structures of the wild type sequence (NC_045512*) predicted by RNAFold and SHAPE Based [26]. We observed that the base pairs predicted in both the cases are comparable. For A26194, A at the 26194th site does not bind with any other nucleotide base before the mutation, forming CAG repeats. For C in C28554, is found in the C–G pair in the wild type sequence in both the methodologies, suggesting that it is very likely that the base pair predicted by RNAfold will occur. When A is mutated to U at the 26194th site, it is in the A-U pair, where U at the 26194th site binds with A at the 26141st site, which might lead to a change in the secondary RNA structure locally.

ORF3a possesses an N-terminal, a transmembrane, and a C-terminal domain folded as 8-strand β-barrel. The mutation T268S was located on the C-tail end of the protein (Figure 7A). The N protein, usually ranging from 419 to 422 amino acids, shows N-terminal domain (NTD), SR-rich motif and C–terminal domain (CTD). The NTD acts as an RNA binding domain, whereas CTD acts as both, dimerization domain and RNA binding domain. Mutation S194L was found to be located in the serine-arginine (SR)-rich motif of N protein. 

## 3. Discussion

As observed, reports and publications highlight diverse aspects of the COVID-19 pandemic inclusive of the SARS-CoV-2 genome architecture, clinical data, disease severity, and outcome; they have also highlighted the need for integrative analysis for enhanced and comprehensive understanding. This study aimed to understand and elucidate the plausible relationship between the mutations and disease severity, as well as to investigate the structural alterations in the protein/RNA produced by these SARS-CoV-2 mutations in the Indian population during the early stages of the pandemic. Understanding the pathogenesis and spread of viruses requires potential association of genetic differences with disease severity. Thus, we first looked at the spectrum of clinical manifestations in the 196 patients’ cohort in the study (Table 1). The patient group was subdivided into the Recovered (R) and Dead (D), which highlighted significant (*p* < 0.05) differences between the age of patients who recovered (32–64 years) and those who succumbed (55–70 years) to COVID-19. Less than half of the recovered patients required respiratory support (44%), while nearly all the patients who died were on respiratory support (100%) (Figure 1A). We observed a relatively shorter duration of ventilator support for patients with severe outcomes. This is probably due to the disease severity wherein although patients were given ventilator support, the revival from the severe condition was lower compared to the patients with moderate disease severity. Multiple comorbidities are associated with the severity of COVID-19 disease progression and cardiovascular comorbid conditions [27]. This was also evident in our cohort—the recovered and dead patients varied significantly in the presence of co-morbidities like cardiovascular conditions and hypertension (Table 1). Patients with type 2 diabetes are known to have increased severity of COVID-19 due to poorer blood glucose control [28]. However, recovered patients with moderate COVID-19 disease in our cohort had this co-morbidity as well (Figure 1B).

A significant number of patients in the R group showed fever and sore throat symptoms (Table 1). However, breathlessness was seen mostly in the D group patients (72%) as evident by the requirement of respiratory support by all the patients who did not survive (Figure 1A). The patients that were treated with antivirals and steroids showed no significant difference in both the groups. However, a significant proportion of patients treated with hydroxychloroquine (HCQ) belonged to the recovered group. HCQ was authorized as an emergency drug from the FDA for COVID-19 on 28March 2020. Although it is important to note that based on early experimental evidence and widespread use of HCQ seen during the firstwave of COVID-19 in India, the role of HCQ in preventing or treating COVID-19 was ruled out subsequently [29]. 

The fitness of the virus strain and its transmission depends on the adaptive mutations that it acquires with time. Within our samples, we identified three frequent lineages of SARS-CoV-2: B.1, B.1.36, and B.6.Lineage B.1.36 was first reported from Saudi Arabia in February 2020 and later circulated in India. Lineage B.6 (belonging to Nextstrain clade 19A) was a major contributor of SARS-CoV-2 during the early pandemic transmission in Malaysia and is known to have spread from there to India [30]. Of the most frequent mutations seen in our samples, P314L and D614G, seen in more than 60% of the population, is known to form a haplotype along with mutations C3073T and C241T (Table 2) [31]. 

Based on the association study between mutations vis-a-vis disease severity and mortality, we selected the mutations, A26194T, C28854T, and C25611A for structural analysis (Table 3). The presence of these mutations was higher in severe and fatal cases. Of these three mutations, the C28854T (S194L) variation has already been reported to be associated with highly severe cases [21]. Nearly 92% and 72% of the cases with mutation A26194T and C26511A were severe compared to an average of 50% severity rate in our cohort. Whereas 38% and 20% of the cases with the mutation A26194T and C28854T were fatal compared to an average mortality rate of 10.5%, which highlights that these mutations are positively correlated with COVID-19 severity and mortality.

Upon analyzing these mutations for their effect on the protein and RNA secondary structure, it was observed that minor changes in the polarity of the A26194T (T268S) were seen. In ORF3a where A26194U occurred, a 6nt loop with CAG repeats was seen before the mutation (Figure 6A); however, when CAG changes to CUG, the loop is converted to the stem loop structure (Figure 6B). For mutation C25611A, a complete change of secondary structure of RNA was observed, resulting in a decrease in positional entropy (blue: high positional entropy, red: low positional entropy) and an increase in base pair probability of the nucleotide bases (blue: low base pair probability, red: high base pair probability) (Appendix A). A study on ORF3a of both SARS-CoV and SARS-CoV-2 identified distinct functional domains associated with virulence, infectivity, and virus release [32]. Our study showed that nearly 59% of the samples with this mutation had a severe COVID-19 disease outcome and ~20% of the infections ended in mortality (Table 3). For mutation C28854T (S194L), a change of secondary structure of RNA was observed resulting in an increase in positional entropy (blue: high positional entropy, red: low positional entropy) and a decrease in base pair probability of the nucleotide bases (blue: low base pair probability, red: high base pair probability) (Appendix A). S194L mutation was found in the SR-rich motif of N protein (Figure 7B), which is shown to be important for virus replication [33]. Although changes are observed in the RNA secondary structures using RNAfold and other structure prediction tools, future additional experimental evidence would be important to understand and elucidate its functional role.

Structural changes in the protein caused by mutations might be responsible for the transmission of SARS-CoV-2 [34]. Our study focuses on analyzing the structural changes at both RNA and protein levels that might be associated with changes in the function and structure of the RNA and protein. Previous studies observed mutation in Orf3a associated with high mortality and infection [35]. Some studies also suggest that the mutation in Orf3a leads to carrying a mutation in the Spike gene, which facilitates viral entry in multiple hosts by interacting with ACE-2 receptors [35,36]. 

The results reported here have made an effort to connect the significance of mutations in modifying the SARS-CoV-2 disease severity and outcome. The study highlights the importance of integrative analysis in which clinical and genetic components of the disease are co-analyzed (Figure 8). 

The clinico-genomic analysis would be extremely useful for enhancing and understanding the association of mutation/s with the clinical sub-phenotype. These mutations have potential to be used in future as part of genetic screening to identify susceptible and protected groups of SARS-CoV-2 infected patients. This will help to proactively identify the patient sub-groups who may require priority healthcare to minimize disease severity and possible mortality. If this integrative framework can be sustained over the future, it would have multi-dimensional benefits inclusive of better usage of healthcare infrastructure and manage strain on the existing medical support system. In combination, it may enhance our future pandemic preparedness [33,37].

## 4. Materials and Methods

### 4.1. Sample Collection and Sequencing

The study was conducted by the CSIR-Institute of Genomics and Integrative Biology (CSIR-IGIB) and the International Institute of Information Technology, Hyderabad (IIITH) in collaboration with MAX Hospital, Delhi, India. Ethical clearance for the study was obtained from the Institutional Ethics Committee at the IGIB and the Max Hospital, respectively. A total of 196 patients with confirmed COVID-19 positive status, based on RT–PCR results, who were hospitalized in MAX Hospital, were enrolled in the study. The nasopharyngeal and/or throat swabs along with a sputum sample were collected by the paramedical staff at the MAX Hospital on the day of reporting to the hospital. The swabs were preserved in a vial containing 3 mL of Viral Transport Media (VTM) (HiViral Transport Kit, HiMedia, Cat. No: MS2760A-50NO), by breaking the applicator’s stick and sealing the tube tightly. The tube was then vortexed for 2 min to allow the dissolution of the sample into the VTM solution, followed by brief centrifugation and was allowed to settle for some time before processing. For sputum samples, 200μLof sputum was added to 200μLof Sputum Liquefaction reagent (Cat. 28289 from Norgen Biotek Corp, Thorold, ON, Canada), thoroughly mixed, and incubated at 37°C for 10 min. In-house sequencing and data analysis for the 196 SARS-CoV-2 genomes was performed using Oxford Nanopore Sequencing and Illumina-Miseq platforms and the pipelines used were the same as detailed in our prior studies [38,39].

### 4.2. Phylogenetic and Mutation Analysis

All the sequences of the COVID-19 positive samples were aligned to the NC_045512 reference genome using the MAFFT [40] multiple alignment tool. The aligned sequences were trimmed to remove gaps and a phylogenetic tree was generated using the default model of the IQ-TREE tool [41]. The tree was visualized using the Figtree tool [12]. The lollipop plot is generated in RStudio using g3viz, rtracklayer, and trackViewer packages, followed by data visualization using the ggplot2 package [42,43]. All the figures were updated using Inkscape software [44].

### 4.3. Statistical Analysis

All the data were summarized using descriptive statistics, wherein continuous variables are presented as a median or interquartile range, and categorical variables are presented as percentages or proportions. We used the Mann–Whitney *U* test and Chi–square tests to compare the differences, wherever appropriate.

### 4.4. Mutation Association Study

To analyze the association of mutations with disease severity, the patients were first stratified according to the levels of disease severity. As COVID-19 is primarily characterized by respiratory distress, the extent of care provided to a patient with respect to respiratory support and patient mortality were considered as indicators of severity. Considering the levels of respiratory support provided, all the patients were categorized intomild, moderate, and severe. The patients who died (severe) or were under some form of respiratory support (moderate), or whose condition was explicitly mentioned to be severe were categorized into the moderate/severe group. The remaining patients were grouped as mild. 

### 4.5. Data Preparation

The data was encoded as binary vectors with bits set as one corresponding to the mutations detected in the patient’s sample. Each of the mutations is thus a feature. The presence of many rare mutations results in extreme sparsity and a skewed dataset. Hence, the mutations present in less than 5% of the patients were not considered to remove outliers and have a robust analysis. The severity labels were binarized–0 for mild cases and 1 for moderate/severe cases. Mortality labels were also binarized, with one indicating that the patient had died.

### 4.6. Mutation Selection

The correlation of each mutation with severity was analyzed using the Chi–Squared test. The null hypothesis here stipulates that there is no difference between the distributions of mutations and severity. The *p*-values obtained from the Chi–Squared test were used to rank the mutations in descending order of correlations. The exact process is repeated to find the correlations between the mutations and mortality. Note that mortality is a stronger and more objective indicator of the disease severity as it is devoid of human bias involved in assigning the severity labels. The two lists of mutations, i.e., correlated with severity and correlated with mortality in decreasing order, were used to select a set of ‘significant mutations’ for the downstream structural analysis. The mutations satisfying the following conditions were deemed significant:The mutation is in the top 15 most correlated mutations with respect to both disease severity and disease mortality. We only consider mutations that are highly correlated with both severity and mortality as it highlights the significance of the mutation and helps avoid outliers.The mutation has a statistically significant correlation (*p* < 0.05) with either disease severity or mortality.The mutation is positively correlated with severity, i.e., the presence of mutation increases the severity of the disease. We consider a mutation to be positively correlated if the proportion of severe patients with the mutation is higher than the average.

### 4.7. Structural Analysis

#### Orf3a and N Protein Sequence Collection 

We downloaded the Orf3a, spike sequences from the NCBI-virus-database, and amino acid sequences from the Uniprot database. We used these sequences for mutation analyses. 

### 4.8. Secondary Structure Analysis of RNA

To study the effect of mutations, we used the secondary structure prediction tool RNAfold web server [20]. It predicts the minimum free energy (MFE) structures from the RNA sequences (maximum length 10,000) that need to be uploaded in fasta format. The output is the MFE secondary structure and centroid secondary structure of the RNA sequences. We compared the secondary structures of the RNA sequences before and after the mutation and analyzed the structural change, if any, due to the mutations. We used the CFSSP tool to predict the secondary structure of the protein.

### 4.9. Protein Disorder Prediction

In the absence of a single definition of protein disorder, we can describe it as a two-state model where every single residue is either ordered or disordered. We used the PONDR-VSL2 [21] web server to calculate the protein disorder. It is the first tool based on artificial neural networks, specifically designed for the prediction of protein disorder. It gives the predisposition scores on a scale of 0 to 1, where 0 corresponds to the fully ordered residues whereas 1 corresponds to the fully disordered residues. The residues with a predisposition score greater than 0.5 are considered as disordered. A predisposition score ranging from 0.1–0.25 and 0.25–0.5 are considered to be moderately and highly flexible, respectively.

### 4.10. Protein Structural Analysis of Orf3a and N Protein of SARS-CoV-2 

We used the CFSSP tool [25] to study the effect of these mutations on the protein structure. To study the changes in the physio–chemical properties caused by these mutations, we used Innovagen’s peptide calculator (https://pepcalc.com/, accessed on 1 May 2021). It gives information about the molecular weight, net charge at neutral pH, information about solubility in water, and iso-electric point of the peptide.

## Figures and Tables

**Figure 1 pathogens-10-01109-f001:**
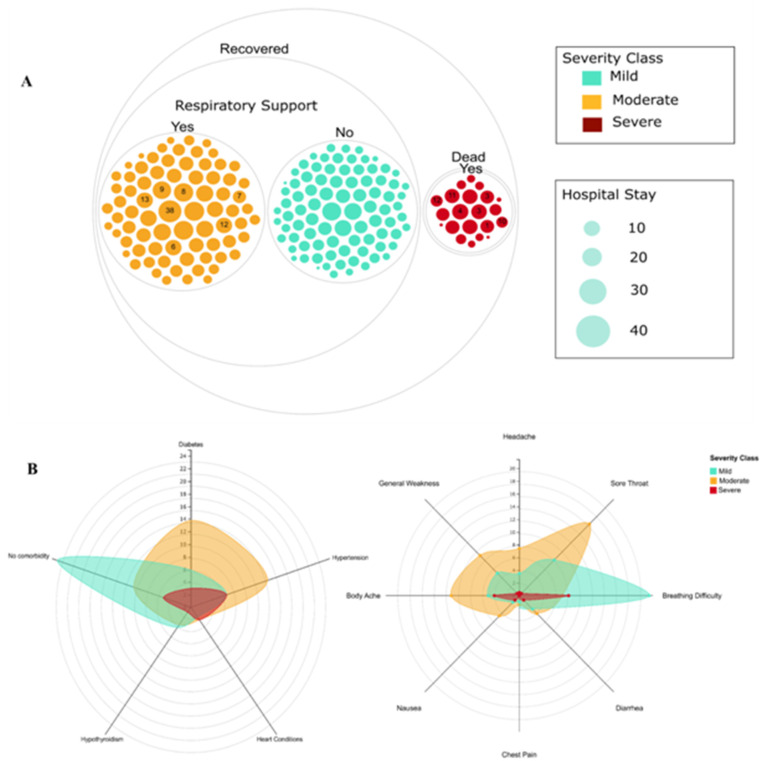
Spectrum of clinical symptoms for the COVID-19 patients in the study. (**A**) Subgrouping of patients based on severity classes: mild (blue)—recovered patients not requiring respiratory support, moderate (orange)—recovered patients requiring respiratory support, and severe (red)—patients who required respiratory support and did not survive. The size of circles represents the duration of hospital stay of each patient and values within the circle represent the duration of ventilator support. (**B**) The spider web plot represents patients’ common co-morbidities, absence of co-morbidity, and clinical symptoms in each severity class.

**Figure 2 pathogens-10-01109-f002:**
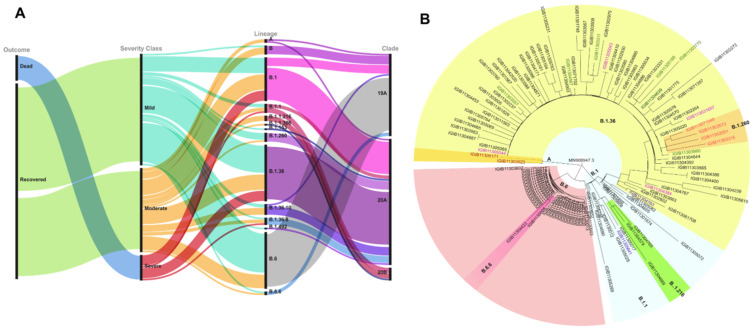
Phylogenetic classification of SARS-CoV-2 isolates. (**A**) Alluvial plot showing the phylogenetic classification of SARS-CoV-2 isolates based on the three severity categories of mild, moderate, and severe. (**B**) Phylogenetic tree of sequences based on PANGOLIN lineage classification.

**Figure 3 pathogens-10-01109-f003:**
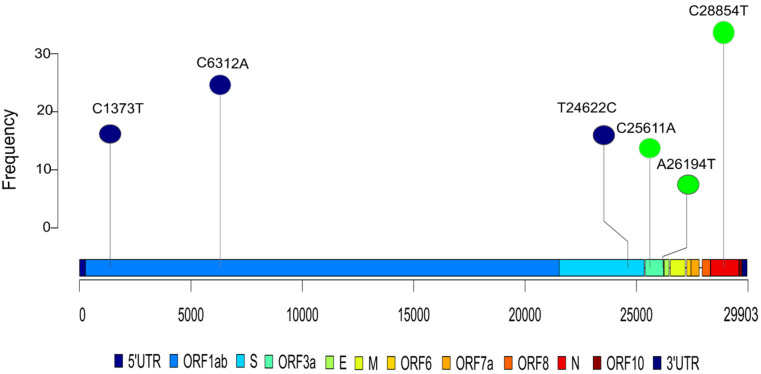
Lollipop plot highlighting the distribution of the associated mutations. It shows the top three positively correlated (in green) and negatively correlated (in blue) mutations significantly associated with severity/mortality. They have been anchored to the genomic region of the SARS-CoV-2.

**Figure 4 pathogens-10-01109-f004:**
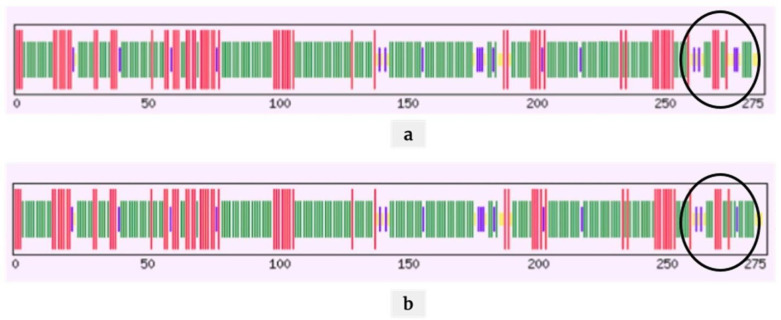
Secondary structural analyses of Orf3a protein for the mutation T268S. Before the mutation (**a**), there were more stem loops (green) and after the mutation (**b**), there were more coils (yellow) in the secondary structure.

**Figure 5 pathogens-10-01109-f005:**
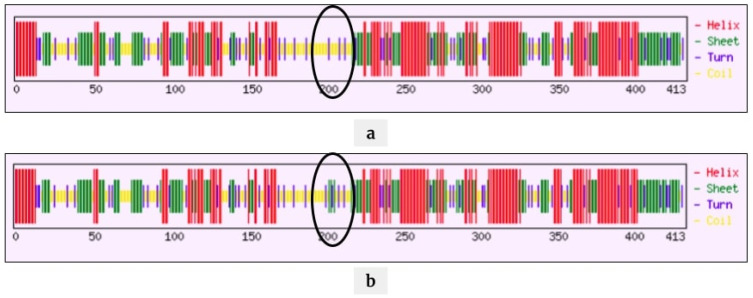
Secondary structural analyses of N protein for the mutation S194L. Before the mutation (**a**), there were more coils (yellow) and after the mutation (**b**), there were more sheets (green) and turns (blue) in the secondary structure of N protein.

**Figure 6 pathogens-10-01109-f006:**
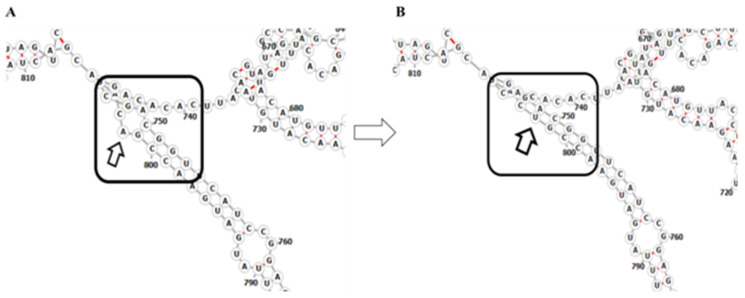
RNA secondary structure of SARS-CoV2 Orf3a. (**A**) 6nt loop in the ORF3A region when A is present at the 26,194th site, A at the 26,141st site (26,194th–26,141st) does not bind with any other residue. (**B**) When A is mutated to T, A present at the 26,141st site binds with T. Here, the 802nd site corresponds to the 26,194th site and the 749th site corresponds to the 26,141st site.

**Figure 7 pathogens-10-01109-f007:**
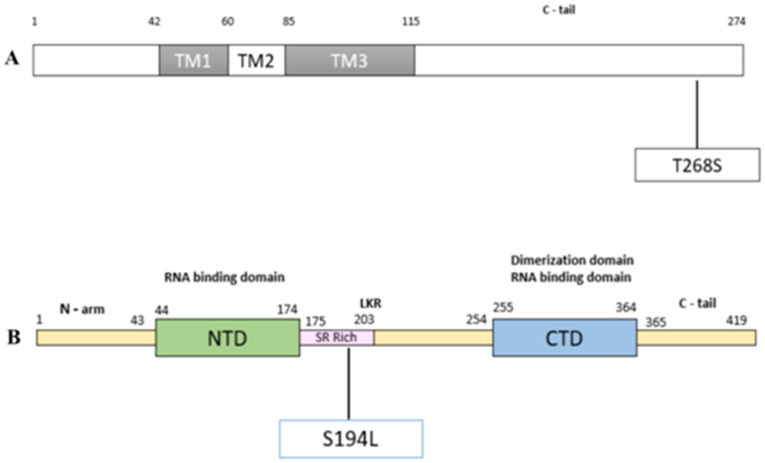
Domain organisation of Orf3a and N protein. (**A**) represents domains of Orf3a protein where mutation T268S is seen on the C-tail end of the Orf3a protein, and (**B**) represents the domains of N protein near mutation S194L, which lies in a serine–arginine (SR) rich motif of N protein.

**Figure 8 pathogens-10-01109-f008:**
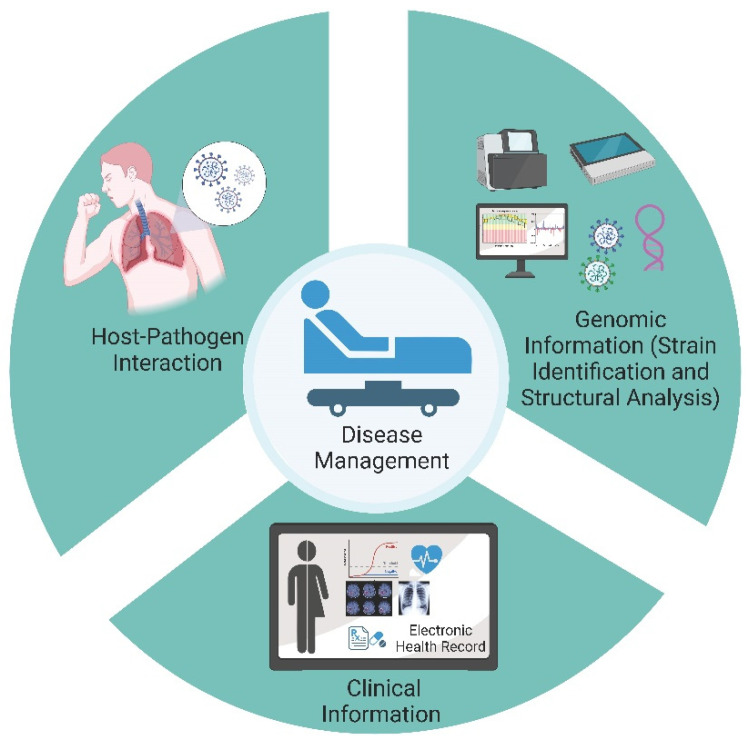
Integrative analysis of Clinical-Genomic-Structure analysis. The figure highlights the benefits of enhanced understanding and elucidation of the host-pathogen interaction through integrative analysis of clinical phenotype, genomics and computational (Structure) analysis. This is more applicable for SARS-CoV-2 with new Variants of Concern leading to surges in different parts of the world.

**Table 1 pathogens-10-01109-t001:** Clinical summary of the SARS-CoV-2 hospital-admitted patients.

Parameter	Total (*N* = 196)	Recovered (*N* = 174)	Dead (*N* = 22)	*p* Value
**Age**	54(36–65)	52(32–64)	65(55–70)	**<0.001** ^a^
**Gender F/M**	58/138	52/122	6/16	0.8 ^b^
**Signs & Symptoms**				
Fever	150(76%)	137(79%)	13(59%)	**0.04 ^b^**
Cough	93(47%)	86(49%)	7(32%)	0.119 ^b^
Sore Throat	49(25%)	48(28%)	1(5%)	**0.018 ^b^**
Headache	23(12%)	22(13%)	1(5%)	0.266 ^b^
Loss of Taste and Smell	2(1%)	2(11%)	0	-
Breathing Difficulty	72(37%)	56(32%)	16(72%)	**<0.001 ^b^**
Chest Pain	5(2%)	5(3%)	0	-
General Weakness	29(15%)	28(16%)	1(5%)	0.15 ^b^
Body Ache	40(20%)	33(19%)	8(36%)	0.058 ^b^
Diarrhoea	16(8%)	14(8%)	2(9%)	0.866 ^b^
Nausea	14(7%)	12(7%)	2(9%)	0.7 ^b^
**Hospital Stays**	11(7–16)	11(7–15)	12(6–17)	0.71 ^a^
**Respiratory Support**	98(50%)	76(44%)	22(100%)	**<0.001 ^b^**
**Ventilator Support**	17(4–12)	9(7–13)	8(2.5–8.75)	**0.034 ^a^**
**Ct Values**				
E	25.05(21.5–27.5)	25.17(21.6–27.5)	23.47(19.6–27.0)	0.33 ^a^
RdRp	26.40(22.6–29.2)	26.53(22.7–29.5)	25.41(22.1–28.06)	0.211 ^a^
**Comorbidities**				
Diabetes	46(23%)	40(23%)	6(27%)	0.655 ^b^
Hypertension	54(27%)	41(24%)	13(59%)	**<0.001 ^b^**
Heart Conditions	14(7%)	9(5%)	5(23%)	**0.0025 ^b^**
Hypothyroidism	17(9%)	15(9%)	2(9%)	0.941 ^b^
**No Co-morbidities**	80(40.81%)	73(41.9%)	7(31.81%)	0.065 ^b^
**Treatment**				
Antiviral	61(31%)	57(33%)	4(18%)	0.164 ^b^
Steroid	66(34%)	61(35%)	5(23%)	0.248 ^b^
Hydroxychloroquine (HCQ)	93(50%)	93(47%)	5(23%)	**0.006 ^b^**

Data are shown as median (IQR) or n(%). ^a^ Mann Whitney U test. ^b^ Chi^2^test.

**Table 2 pathogens-10-01109-t002:** Topmost frequent mutations in our cohort.

Position/SNP	Gene	Amino Acid Change	Frequency (*n* = 196)
C14408T	ORF1b	P314L	75.0
A23403G	S	D614G	63.8
C18877T	ORF1b	-	54.1
C26735T	M	-	51.5
G25563T	ORF3a	Q57H	51.0
C3037T	ORF1a	-	50.5
G11083T	ORF1a	L3606F	43.9
C22444T	S	-	39.3
C28854T	N	S194L	36.2
C6312A	Nsp3	T2016K	29.1
C28311T	N	P13L	28.1

**Table 3 pathogens-10-01109-t003:** Mutations correlated with disease severity and mortality.

Mutation	*p*-Value (Mortality)	*p*-Value (Severity)	Frequency (*n* = 196)	Locus	Amino Acid Change	Severity Rate	Mortality Rate	Frequency (%)
C631A	0.1203	0.0236	33	Orf1a:122	-	0.3030	0.0303	**16.83**
C1373T	0.0949	**0.0077**	36	Orf1a	C370R	0.2778	0.0278	18.36
C3037T	0.0928	0.1317	99	Orf1a:924	-	0.5758	0.1717	50.51
C6312A	**0.0192**	**0.0243**	57	Orf1ab	T2016K	0.3509	0.0175	29.08
T24622C	**0.0335**	**0.0395**	34	S:1020	-	0.3235	0.0000	17.34
C25611A	0.1340	**0.0158**	29	**Orf3a:74**	-	0.7241	0.2069	14.79
A26194T	**0.0028**	**0.0023**	13	**Orf3a**	**T268S**	0.9231	0.3846	6.63
C28854T	**0.0366**	0.1228	71	**N**	**S194L**	0.5915	0.1972	**36.22**

## Data Availability

All the consensus fasta for the samples included in this study are available at GISAID–EpiCoV (https://www.gisaid.org/, accessed on 30 December 2020) under the submissions having sequence ID identifiers as MaxCov* and IGIB113*. Further inquiries can be directed to the corresponding author.

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
