# Peer review of "Clinico-Genomic Analysis Reveals Mutations Associated with COVID-19 Disease Severity: Possible Modulation by RNA Structure"

_pathogens, 2021, doi:10.3390/pathogens10091109_

Round 1

Reviewer 1 Report

Mehta et. al. performed integrative analysis of the mutations in the genome of SARS-CoV-2 in mild, moderate and severe patients, and identified several genomic mutations that related with the severity and mortality in the cohort they studied. The authors examined the potential impact of these mutations and revealed that they could cause changes on both protein and RNA structures. While the mutation and disease severity correlation provide important information, the structural analysis on protein and RNA levels are superficial and some conclusions could be overstated.

Table 3 lists the mutations correlated with disease severity and mortality. What is the criteria of selecting mutations from Table S1 and S2 for Table 3? T24622C was listed in Table 3 but it was not a common mutation in both of the subgroups. I think Table 3 should show the 8 common mutations. Also, it would be useful to check if these mutations covary with some of the popular mutations, e.g. D614G, or examine the combination of co-variation in severe and mortality groups.

Page 6, line 170, A26194T and C28854T were mentioned as they have high mortality rate. Why C25611A was not mentioned? Its mortality rate is slightly higher than C28854T (0.2069 vs. 0.1972).

Figure 4, are the top and bottom panels swapped? According to the figure caption, the wildtype protein has more sheet structure and the mutant has more coils. The top panel contains more coils (yellow) to me.  

When discussing the impact of mutations on protein structure, the authors should provide more information about the Orf3a and N proteins, e.g. domain organizations. T268S is on the C-term tail of Orf3a protein, and the S194L is on the linker of N protein connecting RNA binding domain and dimerization domain. The mutations may have minimal impact on the overall protein structure, but these linkers could participate in some protein:protein interactions that are related with virus replication and disease severity.

The author performed RNA secondary structure prediction and showed that the point mutations A26194T, C28854T and C25611A lead to RNA secondary structure change. It was claimed “…selective mutations were found to modulate the RNA structure.” However, without experimental evidence, the conclusion could be overstated. Two major problems here.

  • The RNA secondary structure of SARS-CoV-2 genome has been mapped by SHAPE (PMID: 33444546, PMCID: PMC7871767). The authors should predict the impact of the point mutations based on these secondary structures supported by SHAPE data. For example, according to Sun et. al, C28854 is in a G-C pair, and C28854U would change it from G-C to G-U pair, which I don’t think would affect the RNA structure much.
  • The authors used RNAfold web server to predict the RNA structure. It was not clear to me what RNA sequences were used as input for structure prediction. In many cases the input RNA boundary selection affects or even determines the structure prediction. What are the boundaries selection criteria in this study?

When discussing the RNA structures, the authors used “T” instead of “U”. RNA only has U.

When describing the RNA structures, the authors used wrong terminology “sheet” to describe a stem loop structure (or a base paired region).

Figure 5 is difficult to read. It should contain nucleotide numbers and pinpoint the mutation residue. Why are the residues differently colored? Also, please avoid using red and green colors together.

Page 7, for the story flow, the protein disorder analysis paragraph (lines 198-200) should move up and combine with the protein structure prediction paragraph, before the RNA structure analysis.

Author Response

Dear Reviewer,

The authors take this opportunity to acknowledge and thank you for your time towards reviewing the manuscript in a detailed manner and sharing your valuable inputs. We are sure that addressing the suggestions have helped improve the quality of the manuscript in terms of readability and information content.

Best wishes,

Rajesh

# Please see the attachment for detailed response.

Reviewer 2 Report

Mehta and colleagues proposed an interesting research article aimed at elucidating the association between SARS-CoV-2 mutations and COVID-19 severity. For this purpose, the authors collected clinical samples from a wide cohort of mild, moderate and severe COVID-19 patients by performing SARS-CoV-2 genome sequencing. Through both experimental and computational approaches the authors identified three major mutations associated with COVID-19 severe clinical manifestations. Overall, the manuscript is very interesting, however, there are some issues that need clarification and additional analyses should be performed. Below are reported minor/major comments that will improve the quality of the manuscript:
1) Numerous sentences of the Introduction section need supporting references. In addition, the authors a better description of the SARS-CoV-2 pathophysiology and clinical manifestations should be implemented. Indeed, the authors described mainly respiratory manifestations without considering COVID-19 neurological and systemic symptoms. For this purpose, please see:
- PMID: 32979398
- PMID: 32851877
- PMID: 32751841
- PMID: 32834902
2) In Table 1, under the row “Comorbidities” the authors have to add another row indicating the number of COVID-19 patients with “No Comorbidities”. This item is fundamental as the authors have to evaluate statistical differences existing between COVID-19 patients with and without comorbidities;
3) In line with the previous comment, have the authors considered the “Comorbidities” or “No Comorbidities” when they analyzed the severity of diseases according to the different types of mutations? This is a fundamental aspect to avoid bias in the interpretation of data;
4) The authors state: It was observed that patients in the moderate class stayed on ventilator support for an average of 13 days with one patient staying on a ventilator for more than a month, while patients in the severe class had an average stay of  6 days. Is the shorter time on ventilatory therapy observed in patients in the severe class due to patients dying before reaching 13 days? Please clarify this aspect;
5) Please add the group of patients with No comorbidities in the results shown in Figure 1;
6) In Chapter 2.3, the authors should evaluate the association of mutation with disease severity also clustering the COVID-19 patients into the following groups: All comorbidities, no comorbidities, diabetes, hypertension, heart conditions, hypothyroidism;
7) In the Introduction or Discussion section, the authors should better emphasize the importance of NGS for SARS-CoV-2 first characterization, COVID-19 diagnosis and genomic surveillance of variants. For this purpose see: 
- PMID: 33846767
- PMID: 31987001
- PMID: 32973171
- PMID: 32679081
8) Please provide the brand and catalog number of “Sputum Liquefaction reagent”.

Author Response

Dear Reviewer,

The authors take this opportunity to acknowledge and thank you for your time towards reviewing the manuscript in a detailed manner and sharing your valuable inputs. We are sure that addressing the suggestions have helped improve the quality of the manuscript in terms of readability and information content.

Best wishes,

Rajesh

# Please see the attachment for the detailed response.

Round 2

Reviewer 2 Report

The authors well addressed all my comments related to the additional analyses on the "no-comorbidities" group. Some amendments are still required in the Introduction and Discussion sections that should be better described and referenced (See previous comments).

Author Response

The authors well addressed all my comments related to the additional analyses on the "no-comorbidities" group. Some amendments are still required in the Introduction and Discussion sections that should be better described and referenced (See previous comments).

We would like to thank the reviewer for their suggestions. We have added a paragraph in the introduction taking into consideration the articles suggested by the reviewers.

Various diagnostics methods, apart from conventional methods, have been developed to ascertain SARS-CoV-2 infected patients to effectively limit the risk of virus transmission [13]. This includes CRISPR-Cas9 based detection of the wild-type SARS-CoV-2 in general as well as specific variants, for example Alpha variant using RAY (Rapid variant AssaY). It is a paper-strip based platform to identify mutational signatures of the coronavirus variants in a sample. Genome sequencing based identification of the signature mutations associated with the VOC is again the enabler towards development of such diagnostic methods [14].  Genome sequencing also plays a pivotal role in controlling the progression of the pandemic by giving us insights on the viral genome architecture and dynamics. This includes the earlier incidences of the viral entering into a geographical region. The NGS enabled genome sequencing has been instrumental to identify the viral clades, lineages and the role of non-synonymous mutations in altering the protein structure  [15]. SARS-CoV-2 genome analysis can help us identify novel variants, reinfection cases and also to track the source of infection [16].  Genomic surveillance has been instrumental to conclusively prove the incidences of re-infection. The identified mutations have been studied in detail for their role in immune escape through functional studies to enhance and elucidate the mechanism [17] . Genome sequencing coupled with serological studies have been used for genomic characterization and epidemiology of the emerging SARS-CoV-2 variants vis-a-vis multiple waves of infection. With vaccination breakthroughs being reported worldwide, genomic surveillance has been helpful to detect the variants and identify the mutations associated with the immune escape [18]. A study reported by Luke et al demonstrated how rapid genomic sequencing along with epidemiological data was used to investigate health care workers associated infections and their transmission network. [19]. All of the insights highlighted above have been facilitated by genome sequencing of SARS-CoV-2. But at the same time, it has been possible due to the background clinical and epidemiological information from the hospital admitted patients as well as population-level genomic surveillance. Diagnostic tools and NGS have been of great assistance in  the development of vaccines and therapeutics, by assessing the pattern of global spread, and genetic heterogeneity while the pandemic is ongoing [20]. However, an understanding based on the sequence analysis of evolving viral genomes has crucial implications for both strategic planning in disease prevention, disease progression, and the development of vaccines and therapeutics, while the pandemic is ongoing.

We have also updated the references for the discussion based on the suggestions as follows.

“The results reported here have made an effort to connect the significance of mutations in modifying the SARS-CoV-2 disease severity and outcome. The study highlights the importance of integrative analysis in which clinical and genetic components of the disease are co-analyzed. 

The clinico-genomic analysis would be extremely useful for enhancing and understanding the association of mutation/s with the clinical sub-phenotype. These mutations have potential to be used in future as part of genetic screening to identify susceptible and protected groups of SARS-CoV-2 infected patients. This will help to proactively identify the patient sub-groups who may require priority healthcare to minimize disease severity and possible mortality. If this integrative framework can be sustained over the future, it would have multi-dimensional benefits inclusive better usage of healthcare infrastructure and manage strain on the existing medical support system. In combination it may enhance our future pandemic preparedness [33], [37].”